# Feasibility and acceptability of offering breast cancer risk assessment to general population women aged 30–39 years: a mixed-methods study protocol

Sarah Hindmarch [1], Sacha J Howell,[2] Juliet A Usher-Smith [3]
Louise Gorman,[4] D Gareth Evans,[5] David P French [1]

For numbered affiliations see end of article.

**Correspondence to**
Dr Sarah Hindmarch;
sarah.hindmarch@manchester.ac.uk

## ABSTRACT

**Introduction** Breast cancer incidence starts to increase exponentially when women reach 30–39 years, hence before they are eligible for breast cancer screening. The introduction of breast cancer risk assessment for this age group could lead to those at higher risk receiving benefits of earlier screening and preventive strategies. Currently, risk assessment is limited to women with a family history of breast cancer only. The Breast CANcer Risk Assessment in Younger women (BCAN-RAY) study is evaluating a comprehensive breast cancer risk assessment strategy for women aged 30–39 years incorporating a questionnaire of breast cancer risk factors, low-dose mammography to assess breast density and polygenic risk. This study will assess the feasibility and acceptability of the BCAN-RAY risk assessment strategy.

**Methods and analysis** This study involves women undergoing risk assessment as part of the BCAN-RAY case-control study (n=750). They will be aged 30–39 years without a strong family history of breast cancer and invited to participate via general practice. A comparison of uptake rates by socioeconomic status and ethnicity between women who participated in the BCAN-RAY study and women who declined participation will be conducted. All participants will be asked to complete self-report questionnaires to assess key potential harms including increased state anxiety (State Trait Anxiety Inventory), cancer worry (Lerman Cancer Worry Scale) and satisfaction with the decision to participate (Decision Regret Scale), alongside potential benefits such as feeling more informed about breast cancer risk. A subsample of approximately 24 women (12 at average risk and 12 at increased risk) will additionally participate in semistructured interviews to understand the acceptability of the risk assessment strategy and identify any changes needed to it to increase uptake.

**Ethics and dissemination** Ethical approval was granted by North West—Greater Manchester West Research Ethics Committee (reference: 22/NW/0268). Study results will be disseminated through peer-reviewed journals, conference presentations and charitable organisations.

**Trial registration number** NCT05305963.

## INTRODUCTION

Breast cancer is the most common cancer diagnosed worldwide for women, with increasing incidence rates observed in premenopausal women in recent years.[1 2] This is concerning as breast cancer is more frequently lethal in younger women than in those diagnosed aged over 50 years (10-year survival aged <40 years at diagnosis 70% vs 87% in those >50 years).[3] This is due to a combination of factors, notably later stage at presentation and a greater proportion of women developing more aggressive breast cancer subtypes.[4–6] Breast cancer is the leading cause of death in women aged 35–50 years in the UK.[7] Therefore, there is a pressing need to identify younger women at increased risk of developing breast cancer so they can be offered screening and preventive strategies.[8]

Assessment of an individual's breast cancer risk is one proposed approach for identifying young women eligible for screening and preventive strategies.[9] In the UK, a strong

---

### STRENGTHS AND LIMITATIONS OF THIS STUDY

⇒ This is the first study to examine the feasibility and acceptability of comprehensive breast cancer risk assessment for general population women aged 30–39 years.
⇒ This study uses a mixed-methods design; the combination of qualitative and quantitative data will result in a more comprehensive understanding of the processes affecting implementation.
⇒ Outcome measures assessing potential harms and benefits of participating in breast cancer risk assessment will be collected at three time points, allowing for assessment of short-term and long-term effects.
⇒ The quality and completeness of ethnicity data across general practices may be suboptimal for the planned analyses.
⇒ As this is a feasibility study, no information about the effectiveness of breast cancer risk assessment will be provided.

family history of breast cancer or a known high-risk genetic variant in a close relative is the only criteria by which women aged under 50 years can access screening and preventive strategies prior to a diagnosis of breast cancer.[10] However, at least 65% of women who develop breast cancer before the age of 50 years do not have such a family history and are not currently identified as being at increased risk.[3 11]

The reliance on family history belies the progress over recent decades in the identification of additional breast cancer risk factors including those related to reproductive and hormonal history, alcohol consumption, polygenic risk scores and mammographic density. These additional factors have been incorporated into risk prediction models, resulting in improved discrimination across all age groups.[12–15] In the UK, the Predicting Risk of Cancer at Screening (PROCAS) study confirmed it was possible to accurately estimate a woman's individual risk of developing breast cancer at the time of mammographic screening using a self-reported questionnaire of breast cancer risk factors and assessment of mammographic density and polygenic risk.[16] Using this comprehensive approach to risk assessment identified 18% of women as being at least moderate risk of developing breast cancer in comparison to only 3.7% using family history alone.[17] Therefore, a greater number of women were identified who would be eligible for consideration of screening and preventive strategies.[10] Trials are underway internationally to establish the potential effectiveness of risk-based screening strategies for women attending breast cancer screening programmes over the age of 40 years.[18 19] However, inclusion of breast cancer risk assessment at the time of national mammographic screening programmes will miss younger women eligible for screening and preventive strategies. Therefore, the introduction of comprehensive breast cancer risk assessment from an earlier age is currently being considered.

A recent review determined that breast cancer risk assessment for women under 50 years currently satisfies many of the key principles for screening.[20] However, uncertainties remain with respect to the optimal strategy for implementation and potential impact of the invitation process on health inequalities. The Breast CANcer Risk Assessment in Younger women (BCAN-RAY) case–control study (NCT05305963) aims to evaluate a comprehensive breast cancer risk assessment strategy among a diverse ethnic and socioeconomic population of women aged 30–39 years without a strong family history of breast cancer.[21] The BCAN-RAY study aims to primarily assess the impact of mammographic density on breast cancer risk in this age group. To address this, we have developed a low-dose mammogram technique which uses 1/10th or less of the radiation dose of a full-dose screening mammogram making it safer. Furthermore, an automated method of analysis not requiring radiologist review will be used, removing the risk of unnecessary recall for additional imaging. This approach has been shown to be accurate in younger women.[22]

The risk assessment strategy thereby consists of a questionnaire of breast cancer risk factors, low-dose mammography to measure mammographic density and a saliva sample to assess polygenic risk and the presence of pathogenic variants in high and moderate-risk genes. The breast cancer risk assessment strategy adopted in the BCAN-RAY study is herein referred to as the BCAN-RAY approach. Women with a strong family history of breast cancer are ineligible to participate because they can access screening and preventive strategies through referral to Family History, Risk and Prevention Clinics (FHRPCs). Women identified as being at increased risk will be offered an appointment at a FHRPC to discuss their risk result further and potential management options. Options in the UK include access to breast screening from the age of 40 years (if 10-year risk reaches 3% by 40) and preventive strategies such as weight loss or weight gain prevention interventions and risk-reducing medication. Uptake of these screening and preventive strategies by younger women has the potential to facilitate earlier detection of breast cancer and reduce breast cancer mortality.[9]

In line with the MRC Framework for Developing and Evaluating Complex Interventions,[23] it is imperative to assess the feasibility of the BCAN-RAY approach in order to inform future decisions about implementation. One key consideration is a need to assess whether the invitation process exacerbates health inequalities through lower recruitment of ethnic minority populations and women from low socioeconomic backgrounds. Previous efforts to implement risk assessment at the time of mammographic screening have demonstrated these problems.[24] This is important to consider as addressing ethnic disparities in breast cancer mortality has been recognised as a key research priority.[25]

Second, potential harms and benefits need to be identified. There is now considerable evidence on the effects of providing breast cancer risk estimates to women aged 47–73 years recruited via the National Health Service (NHS) Breast Screening Programme. These data indicate that women subsequently had more accurate perceptions of risk with no evidence of significant adverse effects on anxiety or cancer worry.[26 27] Nevertheless, there is a need to show an absence of adverse effects when setting up a new programme with younger women for several reasons. First, one might expect more acute distress among younger women at increased risk as the result may be more unexpected because of a lack of family history of the disease, suggesting anxiety and cancer worry are important outcomes to assess. Second, due to the potential implications of being identified as at increased risk for younger women in terms of reproductive decision-making, a possible harm could be that participants experience remorse or distress over their decision to take part in breast cancer risk assessment. In terms of benefits, it is anticipated that women will feel more informed about breast cancer risk as a result of participation which will enable them to make informed choices about subsequent risk management options.

Finally, it is important to consider acceptability of the BCAN-RAY approach to women aged 30–39 years to optimise the likelihood of future implementation being successful. If the processes of invitation, risk assessment and feedback are unacceptable, then the potential benefits will not be realised. For this study, acceptability is defined as the extent to which women receiving breast cancer risk assessment consider it to be appropriate, based on experienced cognitive and emotional responses to participating in risk assessment, in line with an evidence-based framework of acceptability.[28]

We have previously conducted a qualitative study with women aged 30–39 years which suggested that undergoing breast cancer risk assessment was acceptable in principle.[29] However, risk assessment was presented as a hypothetical prospect in that study so how women may respond once they have experienced it and any changes required to increase engagement and uptake remain unknown.

This study aims to examine the feasibility and acceptability of a strategy to offer breast cancer risk assessment to women aged 30–39 years in a diverse ethnic and socioeconomic geographical region. A mixed-methods approach will be adopted in order to capitalise on the strengths of both quantitative and qualitative methods, resulting in a more comprehensive understanding of the processes affecting implementation.[30] Specific objectives of this study are to:

a. Examine uptake rates according to socioeconomic status and ethnicity to determine impact of the invitation process on health inequalities.
b. Identify potential harms and benefits of participation in breast cancer risk assessment.
c. Understand the acceptability of the BCAN-RAY approach.

## METHODS
### Design
BCAN-RAY is a case–control study.[21] Approximately 1000 women will be recruited between May 2023 and May 2025, 250 women diagnosed with breast cancer when they were aged 30–39 years (cases) and 750 controls currently aged 30–39 years without a strong family history of breast cancer. The present feasibility study involves the control participants only and uses three different analyses to address the three objectives.

### Health inequalities assessment
A between-subjects comparison will be made between women who participated in the BCAN-RAY study and women who declined participation according to socioeconomic status and ethnicity.

### Identification of potential harms and benefits
Quantitative questionnaires will be administered to each woman at three time points; baseline, 6 weeks post risk feedback and 6 months post risk feedback. A between-subjects comparison will be made between average and

---

### Box 1  Study exclusion criteria

⇒ Strong family history of breast cancer defined as a first-degree relative diagnosed with breast cancer under the age of 50 or two or more second-degree relatives diagnosed with breast cancer at any age.
⇒ Already under follow-up in a breast cancer family history clinic or have a known mutation in a moderate or high-risk breast cancer gene.
⇒ Any prior malignancy (excluding non-melanoma skin cancer).
⇒ Had a double mastectomy (both breasts removed).
⇒ Breast implants or breast augmentation surgery.
⇒ Currently pregnant.
⇒ Currently breast feeding or stopped breast feeding less than 6 months ago.
⇒ Any condition that would make breast cancer risk assessment inappropriate such as a severe psychiatric or physical illness (assessed by the individual responsible for identifying and inviting women).
⇒ Unable to understand written English.

---

increased risk women for outcomes assessed at multiple time points.

### Understanding acceptability
A cross-sectional qualitative design will be adopted employing one-to-one semistructured interviews.

### Setting and participants
All general practices across Greater Manchester have been approached for participation in BCAN-RAY as participant identification centres. An electronic database search will be conducted by each practice to identify women aged 30–39 years predicted to meet eligibility criteria. All potentially eligible women will be invited. We expect to recruit a diverse sample in terms of ethnicity and socioeconomic status given that Greater Manchester has one of the most ethnically diverse populations in the UK in addition to some of the most deprived areas.[31 32] Furthermore, general practices in areas of higher ethnic and socioeconomic diversity will be prioritised during setup. Participants meet BCAN-RAY study inclusion criteria if they are (1) born biologically female, (2) aged 30–39 years and (3) able to provide informed consent. Participants cannot take part if they meet any of the exclusion criteria outlined in box 1. A series of eligibility checks will be conducted which are described in the next section.

### Procedure
#### BCAN-RAY study
Participating general practices will send postal invitations to eligible women. The BCAN-RAY invitation letter will contain a QR code and web-link to access the participant information sheet and instructions directing prospective participants to the risk assessment web-based application. Once participants have consented to the study online, they will be directed to the BCAN-RAY risk factors questionnaire based on the Tyrer-Cuzick algorithm.[33] Participants will be able to answer part of the questionnaire, save and return to it at a later date. If a participant does not have

access to the internet or is having difficulty completing the questionnaire, they can provide their answers via telephone to the study team who will manually input the participants' responses into the web-based application. If a strong family history of breast cancer (as defined in box 1) is identified during completion of the risk factors questionnaire, participants will be referred back to their general practitioner (GP) for FHRPC referral and their participation in the BCAN-RAY study will end. Following submission of consent and the risk factors questionnaire, participants will be contacted by telephone or email to arrange the risk assessment appointment which will take place at the Nightingale Centre, part of the Manchester University NHS Foundation Trust. Before an appointment is offered, eligibility to take part will be checked by a member of the study team using an eligibility checklist based on self-report. Women who meet any of the exclusion criteria will be withdrawn from the study. Before the appointment, participants will be sent a saliva sample collection tube in the post and asked to bring the saliva sample along to the appointment, which will be analysed for polygenic risk score (SNP313) and the presence of pathogenic variants in high and moderate-risk genes. At the appointment, a final eligibility check will be conducted based on self-report in case any of the information provided for completion of the eligibility checklist has changed since the participant completed it. Once eligibility has been confirmed, participants will undergo low-dose mammography (two views of one breast only). Breast density will be calculated using a new technique called predicted Visual Assessment Score (pVAS). pVAS is an automated method of assessing mammograms using artificial intelligence techniques.[22 34] A risk feedback letter will be generated based on the answers participants give in their questionnaire, the results of genetic testing and mammographic density. The risk feedback letter will inform women that they are at 'average' risk (<3% 10-year risk) or 'increased' risk (≥3% 10-year risk). The decision to not provide women with information about the relative impact of each risk component in the risk feedback letters was informed by findings of a qualitative study we conducted with women who matched the intended recipients of the feasibility study.[29] This study investigated information and support needs with respect to breast cancer risk assessment and risk communication and found that information about the factors contributing to risk was perceived as interesting but generally unhelpful when receiving initial notification of the risk result. Instead, information about what would happen next in terms of proactive risk management was considered most important. Each letter, therefore, focuses on explaining the implications of the risk result (see online supplemental file 1). Participants identified as at increased risk will be offered an appointment at a FHRPC to discuss their risk result further with a breast clinician with expertise in risk assessment, screening and prevention. At this appointment, potential management options including earlier access to breast screening and risk-reducing

medication will be discussed. All participants will receive their risk feedback letter within 16 weeks of the risk assessment appointment, along with leaflets providing additional detail on ways of reducing breast cancer risk, signs and symptoms of breast cancer and breast awareness. An updated risk feedback letter will be sent at the end of the study once the magnitude of risk associated with density is determined more accurately in this age group using all case–control subjects. The timeline from the participant's perspective is shown in figure 1.

### Health inequalities assessment

GPs from participating general practices will extract self-reported ethnicity (where available) and deprivation information based on residential postcode for all women invited to take part in the BCAN-RAY study so that these characteristics can be compared between those who participated in the study and those who declined participation. They will provide this aggregated, non-identifiable data to the research team. No personally identifiable data will be shared with the research team as we predict the majority of women invited will not consent to the study. A member of the research team will then extract the same information from the BCAN-RAY study database for all participants.

### Identification of potential harms and benefits

Once participants have submitted the risk factors questionnaire on the web-based application, they will be directed to complete the baseline harms and benefits questionnaire on Qualtrics (https://www.qualtrics.com/uk/). If the baseline questionnaire has not been completed by the time a member of the study team rings the participant to arrange their risk assessment appointment, a reminder to do so will be enclosed with their appointment confirmation letter. Any remaining non-completers will be asked to complete the questionnaire online or via paper in the waiting room before their risk assessment appointment.

The same women will be asked to complete follow-up questionnaires 6 weeks and 6 months after they have received their risk feedback. Women will be asked to input their unique BCAN-RAY study ID and their date of birth at the beginning of each questionnaire to ensure responses can be linked. Participants are able to request paper copies of the follow-up questionnaires to be sent to them via post if preferred. The data recorded on paper copies of all questionnaires will be manually inputted into the Qualtrics platform by a member of the study team. If the follow-up questionnaires have not been completed by 2 weeks after the initial invitations, a reminder to complete the questionnaire will be sent via email or letter.

### Understanding acceptability

A purposive sample of average and increased risk women who complete the baseline questionnaire and have agreed to be contacted will be sent an invitation to participate in a semistructured interview. Demographic characteristics and responses to questionnaires will guide sampling

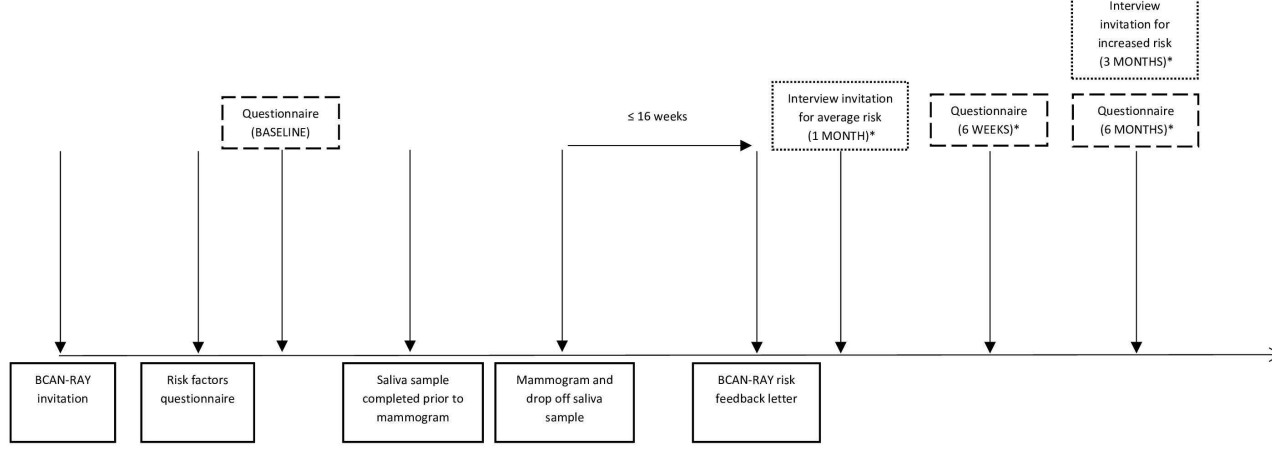

**Figure 1** Timeline of feasibility study integrated with BCAN-RAY. *Duration from risk feedback letter. BCAN-RAY, Breast CANcer Risk Assessment in Younger women.

to allow variation in ethnicity, socioeconomic status and anxiety levels of participants. Average risk women will be invited for interview approximately 1 month after receiving their risk feedback letter. Increased risk women will be invited for interview approximately 3 months after receiving their risk feedback letter. This gives women at increased risk the chance to explore extra screening options or medications prior to the interview and minimises any influence participating in the interview may have on decision-making. We will aim to recruit up to 24 women to these interviews (up to 12 women in each risk category). If no response is received following the initial invitation, a second invitation will be sent approximately 3–4 weeks later.

Interviews will last approximately 40–60 min and will be conducted face to face or over the telephone according to each participant's preference. For face-to-face interviews, written consent will be obtained. For telephone interviews, verbal consent will be obtained over the telephone before the interview begins and recorded in a separate audio file. Interviews will be audio recorded and transcribed verbatim using an accredited transcription company. Participants will be compensated for their time with a £20 shopping voucher.

## Measures
### Health inequalities assessment
Residential postcode, a proxy measure of socioeconomic status, will be converted into deprivation deciles using the Index of Multiple Deprivation (IMD), a measure of relative deprivation for small areas in England.[35] Where available, ethnicity data will be mapped onto the five high-level ethnic categories used in the 2021 Census for England (white, Asian/Asian British, black/African/Caribbean/black British, mixed/multiple and other ethnic group), in line with the current ethnicity harmonised standard.[36] Missing data will be captured under two additional categories of refusal to provide information about ethnic group and no data available.

### Identification of potential harms and benefits
The self-reported measures of potential harms and benefits of participation in breast cancer risk assessment to be completed by participants are shown in table 1. A detailed description of each of these measures is provided in online supplemental file 2. Online supplemental file 3 contains a copy of each questionnaire.

### Understanding acceptability
Topic guide development was informed by the aims of the study and a review of the literature. An initial draft was developed by the lead author, a doctoral student in health psychology with qualitative health services research experience. Feedback on this draft was obtained from public contributors and members of the research team (DF and JU-S) who have research expertise in breast cancer and screening services, primary care and health services research, health psychology and qualitative methods. The content and structure of the topic guide were revised in line with the feedback received. Participants will be asked

**Table 1** Self-reported measures to be assessed, at each of the three time points

| Baseline | 6 weeks post risk feedback | 6 months post risk feedback |
|---|---|---|
| State anxiety[42] | State anxiety[42] | State anxiety[42] |
| Cancer worry[43] | Cancer worry[43] | Cancer worry[43] |
| Risk perception[44] | Risk perception[44] | Risk perception[44] |
| Attitudes towards risk assessment[45] | | Attitudes towards risk assessment[45] |
| | Knowledge* | |
| | Satisfaction with risk feedback information[46] | |
| | | Satisfaction with decision to participate in breast cancer risk assessment[47] |

*Assessed by a measure the research team has created as no validated measure available (see online supplemental file 2 for more information about development of this measure).

about their experience of the risk assessment process including how acceptable they found it, their views on the materials developed for BCAN-RAY, and how the risk assessment process could be improved in terms of delivery/access and provision of information and support (see online supplemental file 4). Furthermore, women will be asked to discuss any actions they have considered and/or made as a result of participating in BCAN-RAY (eg, lifestyle modifications, additional screening and risk-reducing medication).

**Data analysis**
Health inequalities assessment
The $\chi^2$ test will be used to compare uptake rates by ethnicity and socioeconomic status (assessed by IMD deciles) between women who participated in the BCAN-RAY study and women who declined participation. To ensure sufficient instances in each group, IMD deciles will be collapsed into quintiles and ethnicity will be collapsed into six subgroups (white, Asian, black, mixed or multiple, other and missing).

Identification of potential harms and benefits
The main analyses will focus on comparing the responses of the two groups of women provided with different risk estimates (average and increased) for outcomes assessed at multiple time points (ie, anxiety, cancer worry, risk perceptions and attitudes towards breast cancer risk assessment). Analysis of covariance (ANCOVA) will be used, with baseline responses to the same variables, age and IMD deciles as covariates. Analyses will be conducted on all questionnaire measures at 6 weeks and 6 months, with the 6-month state anxiety measure being the primary outcome.

Measures administered at only one time point (knowledge, satisfaction with information received and satisfaction with decision to participate in breast cancer risk assessment) will be compared between the two groups of women provided with different risk estimates (average or increased). ANCOVA will be used, with age and IMD deciles as covariates.

All statistical tests will be two sided and use a significance level of 5%. A 'completer only' analysis strategy will be employed. If drop-out levels are high, the a priori primary outcome (comparison of 6-month outcome scores between average and increased risk groups) will be repeated using a last occasion carried forward approach to missing data as a sensitivity analysis. Statistical analyses will be performed by using SPSS (version 29).

Understanding acceptability
NVivo software will be used to organise the data. Data will be analysed using a manifest-level approach to reflexive thematic analysis.[37 38] Thematic analysis involves examining qualitative data to produce themes that summarise and interpret patterns of results. Initial coding will be deductive based on the structured questions in the topic guide to address the objective of whether the BCAN-RAY approach is acceptable. Inductive methods will then be used to capture additional codes and context to ensure important aspects of the data are not missed. A critical realist approach will be adopted, with the researchers accepting that participants' accounts represent their perception of their reality, which is shaped by and embedded within their cultural context and language.[39] An experiential orientation to data interpretation will be adopted that seeks to stay close to participants' meanings and capture these in ways that might be recognisable to them. The analysis will be conducted by the lead researcher with input from other members of the research team and public contributors.

**Sample size estimation**
Health inequalities assessment
The BCAN-RAY feasibility study aims to recruit approximately 750 women. Based on the results of the latest NHS GP Patient Survey in which 13%–19% of those invited by post aged 25–44 responded,[40] we conservatively expect a response rate of 10%. Therefore, approximately 7500 invitations will be sent. If the response rate is lower than expected, more invitations will be sent until at least 750 women have been recruited. This approach will also yield at least 6750 women who decline participation. Given the geographical spread of the general practices who have provisionally agreed to be involved in the study across different boroughs of Greater Manchester, we expect to recruit a socioeconomically diverse sample (see table 2).

Identification of potential harms and benefits
The sample size for the BCAN-RAY study was based on providing sufficient power to be able to detect an effect

**Table 2** Percentage of lower super output areas in each deprivation decile across the boroughs of Greater Manchester involved in the BCAN-RAY study*

| Deprivation decile† | Location | | | | | |
|---|---|---|---|---|---|---|
| | Trafford | Manchester | Salford | Tameside | Rochdale | Stockport |
| 1–2 (most deprived) | 8.7% | 59.3% | 48.7% | 42.6% | 44.1% | 16.3% |
| 3–4 | 15.9% | 25.8% | 21.4% | 22.7% | 26.1% | 20% |
| 5–6 | 15.2% | 10.7% | 15.3% | 20.6% | 10.4% | 15.3% |
| 7–8 | 25.3% | 3.9% | 7.3% | 12.1% | 15% | 21.6% |
| 9–10 (least deprived) | 34.8% | 0.4% | 7.3% | 2.1% | 4.5% | 26.9% |

*Data sourced from an interactive map created by Greater Manchester Poverty Action.[31]
†Assessed by the Index of Multiple Deprivation 2019.[35]
BCAN-RAY, Breast CANcer Risk Assessment in Younger women.

of breast density, after adjustment for age and body mass index. Therefore, a post hoc analysis was conducted to estimate achieved power with respect to the primary outcome of anxiety at 6 months. Assuming a two-tailed independent samples t-test and follow-up questionnaire responses from 400 average risk women and 100 increased risk women, it is estimated that there will be approximately 76% power to detect a small, standardised difference of d=0.3.

## Understanding acceptability

The sample size for the BCAN-RAY study will provide more than sufficient numbers from which to recruit participants for the acceptability assessment. While we anticipate including up to 24 participants in this component of the study (12 at average risk and 12 at increased risk), the decision to stop recruitment will be guided by the concept of 'information power'. The research team will reflect on the information richness of their dataset throughout data collection to determine when sufficient data has been collected to answer the research question.[41]

## Public involvement

A public involvement group of 11 women aged 30–39 years was established in September 2021 to inform the development of research aimed at identifying young women at increased risk of breast cancer including the BCAN-RAY study. Five women reviewed the study documentation (participant information sheet, consent form, study invite letter, risk feedback letters, baseline and follow-up questionnaires and interview topic guide). The content and structure of documentation were revised in line with the feedback received. Changes included the removal of one question from the knowledge measure as it overlapped considerably with the content of one of the other questions and the addition of breast cancer charity contact information to risk feedback letters. We will continue to involve members of the public involvement group in subsequent stages of the research cycle including analysis of interview data and dissemination.

## Ethics and dissemination

This study was approved by the North West—Greater Manchester West Research Ethics Committee (reference: 22/NW/0268). The study will be performed in accordance with the Declaration of Helsinki, Good Clinical Practice principles and relevant regulations. All participants in BCAN-RAY complete written consent online. All participants will provide informed consent (written if face to face, verbal if over telephone) prior to taking part in an interview. Quantitative study data will be tracked via participant study IDs. Identifying information will be removed from the interview transcripts and participants will be assigned pseudonyms.

We will disseminate our findings through peer-reviewed journals, conference presentations and charitable organisations. At the time of consent for both the BCAN-RAY study and an interview, participants will be asked to indicate whether they wish to receive a summary of findings. A written lay summary will be produced and sent to those who opt to receive this.

## DISCUSSION

The present research aims to provide evidence on the feasibility of a strategy to offer breast cancer risk assessment based on family history, phenotypic risk factors, polygenic risk and mammographic density to women aged 30–39 years. It will provide information about uptake rates, potential harms and benefits of participation, and the acceptability of the risk assessment strategy including novel insight into the experience of low-dose mammography among a population of women not known to be at increased risk of breast cancer.

One key issue that the present research does not cover relates to whether BCAN-RAY is acceptable to healthcare professionals involved in its delivery, which is recognised as an important component of feasibility.[23] We have interviewed and conducted focus groups with primary care professionals to understand their views on involvement in breast cancer risk assessment and management and analysis is ongoing. However, as the optimal strategy for

implementation remains unclear, it is not yet known who would be responsible for the delivery of risk assessment. Future research investigating alternative strategies for implementation ought to consider the views of healthcare personnel involved in delivery to establish likely effects on the healthcare system when implementing risk assessment.

The study will provide valuable information about whether a primary care co-ordinated invitation process is successful at engaging women from diverse socioeconomic and ethnic backgrounds thereby informing the need to consider and evaluate alternative invitation methods prior to further implementation. Furthermore, findings will provide information about the likely harms and benefits of participation in breast cancer risk assessment and identify modifications needed to the risk assessment strategy to increase engagement and uptake in future implementation studies.

Key feasibility issues for implementing risk-stratified screening into routine breast cancer screening have now been identified. This study provides an important first step in assessing the feasibility of introducing comprehensive breast cancer risk assessment for younger women to enable those identified as being at increased risk access to screening and preventive strategies in the absence of a family history of breast cancer.

**Author affiliations**
[1]Manchester Centre for Health Psychology, Division of Psychology and Mental Health, School of Health Sciences, Faculty of Biology, Medicine and Health, The University of Manchester, Manchester, UK
[2]Division of Cancer Sciences, Faculty of Biology, Medicine and Health, University of Manchester, Manchester Academic Health Science Centre, The University of Manchester, Manchester, UK
[3]Primary Care Unit, Department of Public Health and Primary Care, University of Cambridge, Cambridge, UK
[4]NIHR Greater Manchester Patient Safety Research Collaboration, Division of Population Health, Health Services Research & Primary Care, Faculty of Biology, Medicine and Health, The University of Manchester, Manchester, UK
[5]Manchester Academic Health Science Centre, Division of Evolution and Genomic Sciences, School of Biological Sciences, Faculty of Biology, Medicine and Health, The University of Manchester, Manchester, UK

**Acknowledgements** We would like to thank Stephanie Archer who helped with the developmental work and Brian McMillan for advising on ethnicity data collection and reporting in primary care. We also gratefully acknowledge the contributions of our public involvement group.

**Contributors** The BCAN-RAY study was conceived and designed and is being led by SHowell and DGE. Funding for BCAN-RAY was led by SHowell and DGE, with input from JU-S and DF. The feasibility study and participant documentation were designed by SHindmarch, SHowell, JU-S and DF. SHindmarch co-ordinated the involvement of public contributors. The present article was drafted by SHindmarch. DF, SHowell, LG, JU-S and DGE provided feedback on versions of the manuscript. All authors read and approved the final manuscript.

**Funding** This study is sponsored by Manchester University NHS Foundation Trust and funded by grants from Cancer Research UK Alliance for Cancer Early Detection (ref: EDDAMC-2021\100003) and The Christie Charity. The web-based application was developed through charitable donations from the Shine Bright Foundation and Tony Thornley. The low-dose mammogram was developed with a grant from the Medical Research Council's Confidence in Concept funding scheme (2018/19). SHindmarch is funded by a Manchester Cancer Research Centre PhD studentship. DF, DGE and SHowell are supported by the NIHR Manchester Biomedical Research Centre (IS-BRC-1215-20007 and NIHR203308). JU-S is funded by an Advanced Fellowship from the National Institute for Health and Social Care Research (NIHR300861).

**Disclaimer** The views expressed are those of the authors and not necessarily those of the NHS, the NIHR or the Department of Health and Social Care. These funding sources had no role in the design of this study and will not have any role during its execution, analyses, interpretation of the data, or decision to submit results.

**Competing interests** None declared.

**Patient and public involvement** Patients and/or the public were involved in the design, or conduct, or reporting, or dissemination plans of this research. Refer to the Methods section for further details.

**Patient consent for publication** Not applicable.

**Provenance and peer review** Not commissioned; externally peer reviewed.

**ORCID iDs**
Sarah Hindmarch http://orcid.org/0000-0002-9549-1177
Juliet A Usher-Smith http://orcid.org/0000-0002-8501-2531
David P French http://orcid.org/0000-0002-7663-7804

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
