## [Reviewer comments · BMJ Open]

ARTICLE DETAILS

TITLE (PROVISIONAL)	The feasibility and acceptability of offering breast cancer risk assessment to general population women aged 30-39 years: A mixed-methods study protocol
AUTHORS	Hindmarch, Sarah; Howell, Sacha; Usher-Smith, Juliet; Gorman, Louise; Evans, D. Gareth; French, David

VERSION 1 – REVIEW

REVIEWER	Keogh, Louise The University of Melbourne
REVIEW RETURNED	20-Sep-2023

GENERAL COMMENTS	This is a very well written and clearly described research protocol, for an important study on the feasibility of risk assessment for breast cancer for women younger than the current screening cohort. Only minor suggestions for the authors to consider: p9 when the authors say "three different designs" - do they mean three different sets of analyses? It seems to be one (mixed-method) study with separate analyses to be performed. Can the authors confirm that practices will be responsible for assessing whether patients meet the inclusion/exclusion criteria and that all the relevant information will be available to GPs? or is the eligibility checked by the study based on self-report? The exclusion criteria are quite extensive and I can imagine that some practices will not have all the information on, say a new patient. further discussion on how eligibility will be confirmed would be helpful. How will participants be informed about the possible outcomes of the polygenic risk score? i.e. what pre-test counselling will be provided? How will they make an informed decision about whether to take the test? Please explain why women will not be given information about the risk score and mammographic density individually, rather than just the risk range. Given you have this information I can see value in women having the option of having the more detailed information about their risk. If Polygenic risk scores are given, it would also need to be made clear that these results do not provide information about genetic risk for family members. can the authors discuss this issue further?
--

REVIEWER	Smit, Amelia The University of Sydney , Faculty of Medicine and Health
REVIEW RETURNED	12-Oct-2023

GENERAL COMMENTS	This is well written protocol for a mixed-methods study that will generate novel evidence on the feasibility and acceptability of offering risk assessment to the general population between the ages of 30-39. These findings will be crucial to the successful implementation of risk-stratified approaches to breast cancer screening a population scale. I have only two minor comments: Page 8, line 3: at first mention of acceptability it would be useful for the reader to have a brief definition, for example, is it a component in the MRC Framework for Developing and Evaluating Complex Interventions? I think this is important given that acceptability is used in different ways throughout the literature. Page 12, line 18: it would be useful to specify the rationale for extracting data on ethnicity etc for all women invited from the participating practices (presumably it is to compare characteristics between those who consented and decliners)
--

VERSION 1 – AUTHOR RESPONSE

Reviewer 1 comments	Author response	Page and line number
This is a very well written and clearly described research protocol, for an important study on the feasibility of risk assessment for breast cancer for women younger than the current screening cohort.	We are pleased that the reviewer considered the manuscript to be well written and a useful contribution to the body of existing knowledge. We would like to thank the reviewer for providing constructive comments to improve the paper. We have addressed each of their comments, with changes to the text tracked in the accompanying manuscript.	N/A
p9 when the authors say "three different designs" - do they mean three different sets of analyses? It seems to be one (mixed-method) study with separate analyses to be performed.	On reflection, we agree this is clearer so have amended this sentence. It now reads: The present feasibility study involves the control participants only and uses three different analyses designs to address the three objectives.	Pg 9, line 3
Can the authors confirm that practices will be responsible for assessing whether patients meet the inclusion/exclusion criteria and that all the relevant information will be available to GPs? or is the eligibility	Thank you for this suggestion. We agree that providing additional detail on how eligibility will be confirmed would be helpful. A series of eligibility checks will be conducted so the following sentence has been added to the end of the setting and participants section to highlight this: A series of eligibility checks will be conducted which are described in the next section.	Pg 10, lines 6-7 Pg 11, lines 10-23 Pg 12, line 1

checked by the study based on self-report? The exclusion criteria are quite extensive and I can imagine that some practices will not have all the information on, say a new patient. Further discussion on how eligibility will be confirmed would be helpful.	The eligibility checks are now described in the BCAN-RAY procedure section: If a strong family history of breast cancer (as defined in Table 1) is identified during completion of the risk factors questionnaire, participants will be referred back to their GP for FHRPC referral and their participation in the BCAN-RAY study will end. Following submission of consent and the risk factors questionnaire, participants will be contacted by telephone or email to arrange the risk assessment appointment which will take place at the Nightingale Centre, part of the Manchester University NHS Foundation Trust. Before an appointment is offered, eligibility to take part will be checked by a member of the study team over the phone or via email using an eligibility checklist based on self-report. Women who meet any of the exclusion criteria will be withdrawn from the study. Before the appointment, participants will be sent a saliva sample collection tube in the post and asked to bring the saliva sample along to the appointment, which will be analysed for polygenic risk score (SNP313) and the presence of pathogenic variants in high and moderate-risk genes. At the appointment, a final eligibility check will be conducted based on self-report in case any of the information provided in the risk factors questionnaire has changed since the participant completed it. Once eligibility has been confirmed, participants will undergo low-dose mammography (two views of one breast only).	
How will participants be informed about the possible outcomes of the polygenic risk score? i.e. what pre-test counselling will be provided? How will they make an informed decision about whether to take the test?	No pre-test counselling will be provided. The scope of genetic testing i.e., the assessment of both pathological variants in high/moderate risk genes and small genetic changes to derive a polygenic risk score, is described in detail in the participant information sheet. This includes a description of what a polygenic risk score is and how it is calculated is provided in the participant information sheet. This information was reviewed by public contributors and deemed sufficient for participants to be able to make an informed decision about whether to undergo the testing or not. Furthermore, prospective participants are encouraged to contact the research team if they have any questions or concerns about participating in the study.	N/A

Please explain why women will not be given information about the risk score and mammographic density individually, rather than just the risk range. Given you have this information I can see value in women having the option of having the more detailed information about their risk. If Polygenic risk scores are given, it would also need to be made clear that these results do not provide information about genetic risk for family members. Can the authors discuss this issue further?	Thank you for this feedback. The decision to not provide women with information about the relative impact of each risk component in the risk feedback letters was informed by findings of a qualitative study we conducted with women who matched the intended recipients of the feasibility study. This study investigated information and support needs with respect to breast cancer risk assessment and risk communication and found that information about the factors contributing to risk was perceived as interesting but generally unhelpful when receiving initial notification of the risk result. Instead, information about what would happen next in terms of proactive risk management was considered most important. Each letter therefore focuses on explaining the implications of the risk result. We have amended the text to make this clearer: The risk feedback letter will inform women that they are at “average” risk (< 3% 10-year risk) or “increased” risk (≥ 3% 10-year risk). The decision to not provide women with information about the relative impact of each risk component in the risk feedback letters was informed by findings of a qualitative study we conducted with women who matched the intended recipients of the feasibility study (29). This study investigated information and support needs with respect to breast cancer risk assessment and risk communication and found that information about the factors contributing to risk was perceived as interesting but generally unhelpful when receiving initial notification of the risk result. Instead, information about what would happen next in terms of proactive risk management was considered most important. Each letter therefore focuses on explaining the implications of the risk result (see supplementary file 1). Women identified at increased risk will be invited to a risk review appointment to discuss their risk result further with a breast clinician with expertise in risk assessment, screening and prevention. At the risk review appointment more information will be provided including the relative impact of polygenic risk score vs density. This discussion will clarify that polygenic risk scores do not provide information about genetic risk for family members.	Pg 12, lines 8-17
--	---	--------------------------

	All controls will be given feedback about the results of the gene mutation search. Should a pathological variant in a high/moderate risk gene be identified, the participant is informed that they will have the opportunity to discuss the implications of this result for family members at the risk review appointment.	
--	---	--

Reviewer 2 comments	Author response	Page and line number
This is well written protocol for a mixed-methods study that will generate novel evidence on the feasibility and acceptability of offering risk assessment to the general population between the ages of 30-39. These findings will be crucial to the successful implementation of risk-stratified approaches to breast cancer screening a population scale.	We are pleased that the reviewer considered the manuscript to be well written and a useful contribution to the body of existing knowledge. We would like to thank the reviewer for providing constructive comments to improve the paper. We have addressed each of their comments, with changes to the text tracked in the accompanying manuscript.	N/A
Page 8, line 3: at first mention of acceptability it would be useful for the reader to have a brief definition, for example, is it a component in the MRC Framework for Developing and Evaluating Complex Interventions? I think this is important given that acceptability is used in different ways throughout the literature.	Thanks for this comment. Acceptability is a key component of feasibility according to the MRC framework i.e., it is recognised that acceptability is necessary but not sufficient to produce feasibility. However, as this framework does not explicitly define acceptability, we have used the Sekhon approach instead. The text now reads: Finally, it is important to consider acceptability of the BCAN-RAY approach to women aged 30-39 years to optimise the likelihood of future implementation being successful. If the processes of invitation, risk assessment and feedback are unacceptable, then the potential benefits will not be realised. For this study, acceptability is defined as the extent to which women receiving breast cancer risk assessment consider it to be appropriate, based on experienced cognitive and emotional responses to participating	Pg 7, lines 21-22 Pg 8, lines 1-2

	in risk assessment, in line with an evidence-based framework of acceptability (28). References added: 28. Sekhon M, Cartwright M, Francis JJ. Acceptability of healthcare interventions: an overview of reviews and development of a theoretical framework. BMC Health Serv Res. 2017;17(1):88. https://doi.org/10.1186/s12913-017-2031-8	
Page 12, line 18: it would be useful to specify the rationale for extracting data on ethnicity etc for all women invited from the participating practices (presumably it is to compare characteristics between those who consented and decliners)	Thank you for this suggestion. We have included the rationale for extracting this data. The text now reads: GPs from participating general practices will extract self-reported ethnicity (where available) and deprivation information based on residential postcode for all women invited to take part in the BCAN-RAY study so that these characteristics can be compared between those who participate in the study and those who decline participation.	Pg 13, lines 8-9

Additional changes made by authors	Page and line number
Since the manuscript was submitted, there has been a change to the study protocol. The timeframe to invite women identified as being at increased risk of breast cancer to take part in the acceptability interviews has changed from 6 months post risk feedback to 3 months post risk feedback. 3 months will still give ample time for women to have attended the risk consultation and will likely lead to better recollection of the consultation and risk assessment process given less time will have passed. Therefore, Figure 1 has been updated and the following text amended to reflect this change: Increased risk women will be invited for interview 3 6 months after receiving their risk feedback letter.	Figure 1 Pg 14, line 16
The acknowledgements section has been updated as an omission was noted. The text now reads:	Pg 24, lines 18-19

We would like to thank Stephanie Archer who helped with the developmental work and Brian McMillan for advising on ethnicity data collection and reporting in primary care. We also gratefully acknowledge the contributions of our public involvement group.	
Updated the access date for reference 25.	Pg 29, line 11